# How Does Digital Economy Promote the Geographical Agglomeration of Manufacturing Industry?

**Meijuan Wang** [1,2,*] ，**Mingzhi Zhang** [3,*]，**Haiqian Chen** [1] **and Donghua Yu** [1]

1   School of Economics, Shandong University, Jinan 250100, China
2   School of Tourism, Shandong Women's University, Jinan 250000, China
3   School of Economics, Shandong University of Finance and Economics, Jinan 250014, China
*   Correspondence: wangmeijuan900901@foxmail.com (M.W.); liysdu@yeah.net (M.Z.)

**Abstract:** With the acceleration of informatization, the spatial layout of economic activities has gradually shifted from "transportation cost + labor force" to "information + technology". As a new generation of information, the digital economy has a profound impact on the spatial layout of the manufacturing industry. Based on the data of China's listed manufacturing companies from 2001 to 2020, this paper aims to assess the effect of the digital economy on manufacturing agglomeration and identify the transmission mechanism of this effect. The results show the following: (1) The digital economy significantly promotes the geographical agglomeration of the manufacturing industry, which is still valid on the basis of a series of robustness and endogeneity tests. (2) Mechanism analysis shows that the digital economy promotes manufacturing agglomeration by reducing transaction costs, increasing market potential and enhancing knowledge spillover. (3) Heterogeneity analysis shows that the effect is more significant in the samples of large enterprises, high-tech manufacturing, central and western regions, small and medium-sized cities and the west side of the "Hu Huanyong Line", which will greatly help the layout of the manufacturing industry break through the "Hu Huanyong Line" to achieve balanced development. (4) Globalization, localization and human capital play a significant positive moderating role in the process. This paper provides microevidence for the integration of digitalization and industrialization. Furthermore, it has important implications for the formulation of digital economy policy, the high-quality development of the manufacturing industry and the continuous promotion of regional coordinated development.

**Keywords:** digital economy; high-quality economic development; geographical agglomeration of manufacturing industry; China

## 1. Introduction

The report of the 20th National Congress of the Communist Party of China points out that high-quality development is the primary task of building a modern socialist country. The development of the manufacturing industry is related to the lifeblood of the national economy and is the key to high-quality economic development [1,2]. As an important carrier of urban and regional development, the formation, agglomeration and diffusion of manufacturing agglomeration areas directly affect the operational efficiency of the social economy and regional spatial pattern, thereby affecting the overall social and economic development [3]. It has also been proven that manufacturing agglomeration can help form a benign competitive pattern, which is an important way to build a strong manufacturing country and promote the high-quality development of the manufacturing industry [4,5]. At the present stage, a wave of "reindustrialization" has been set off among the world's major industrial countries, which means that the world has officially entered the era of manufacturing competition. The "manufacturing reflux" of developed countries, such as the United States, Japan and the United Kingdom, has appeared one after another, with an increasing degree of agglomeration. However, China's manufacturing industry is facing the

problem of unbalanced and inadequate development, and premature "deindustrialization" has become a new problem [6]. Whether from the Gini coefficient calculated by the gross industrial output value and the number of industrial enterprises or from the location entropy calculated by manufacturing employees, China's manufacturing agglomeration has shown a slow downward trend in recent years (Figure 1), which coincides with the opinions of He and Hu [7].

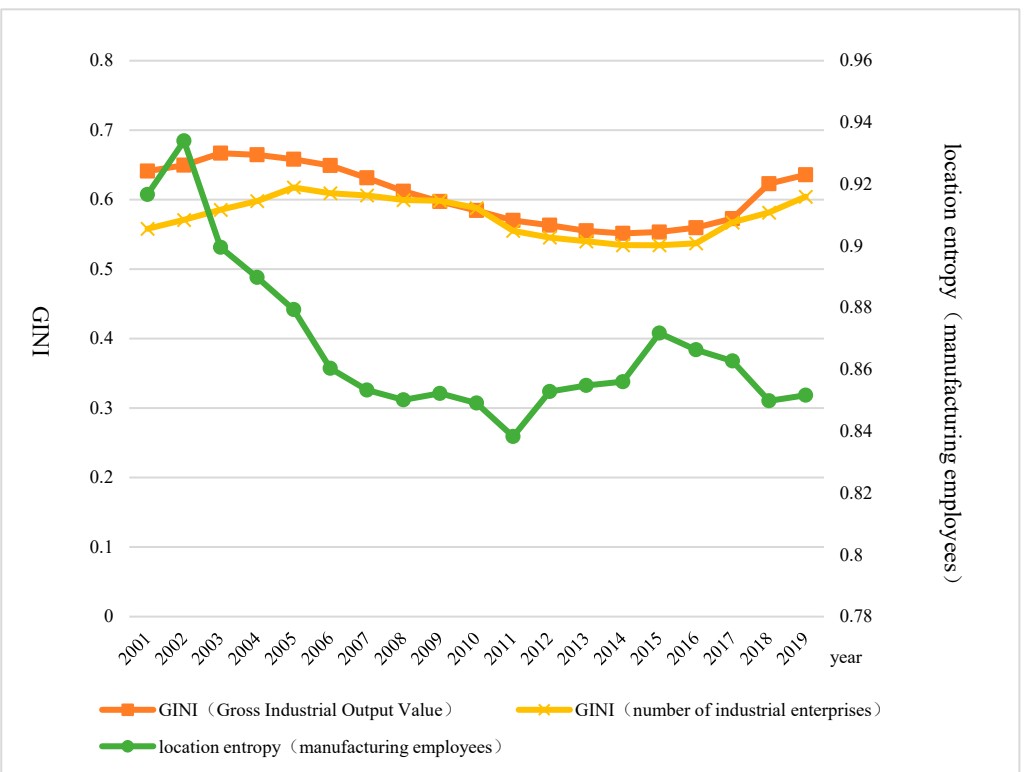

**Figure 1.** Agglomeration characteristics of the manufacturing industry.

As the industry with the most obvious cluster characteristics, the decline of manufacturing agglomeration reflects that the location of industrial economic activities is quietly changing. At the same time, the history of technological development shows that all industrial technological revolutions have promoted the transformation of production modes, and formed a new industrial pattern while promoting industrial upgrading [8]. Especially after the 21st century, the digital economy has become a key factor in reshaping the global competition pattern and the core driving force for high-quality economic development [9,10]. In the process of continuous integration with traditional manufacturing, the digital economy will change the restrictions on factor endowment and geographical location on enterprises, promoting the relocation, migration and agglomeration of enterprises to form a new industrial layout, and thus the "transportation + labor orientation" of industrial location will gradually shift to "information orientation". Therefore, the impact of the digital economy on the location of industrial economic activities (especially manufacturing) has become a hot topic in recent years [11].

Against this background, the present study focuses on addressing these questions: The first question to ask is whether the digital economy, as a new dynamic, can have a significant impact on the geographical agglomeration of China's manufacturing industry; is it helpful to strengthen agglomeration or promote diffusion? There is no consensus, either theoretically or empirically. Secondly, what is the heterogeneity of this effect in terms of regions, cities, industries and enterprises? Thirdly, what is the inner impact mechanism of the digital economy on manufacturing agglomeration? Lastly, what external factors might interfere with this effect? What moderating roles do globalization, localization and human

capital play in this process? Clarifying these questions will further supplement and enrich the literature on the digital economy and provide new ideas and methods for promoting China's manufacturing agglomeration, which has important implications for improving the international competitiveness of the manufacturing industry and building a strong manufacturing country.

The rest of the paper is structured as follows: Section 2 is the literature review. Section 3 presents the research hypothesis. Section 4 introduces the model, variables and data resources. Section 5 interprets the results and the discussion. Section 6 concludes and makes policy recommendations.

## 2. Literature Review

There are three main branches of literature related to this paper: the first focuses on the effects of the digital economy. The effects of the digital economy have been extensively studied in depth, including enterprise innovation at the micro level, industrial upgrading at the meso level, and economic growth, total factor productivity improvement, urban innovation [12] and international trade at the macro level, and most scholars have recognized the positive effects of the digital economy. Digital technology can overcome geographic and technological distances, and facilitate knowledge flows within firms to promote technological innovation [13]. In the meantime, digital transformation improves company performance, and the adoption of new digital processes contributes to greater competitiveness [14]. The digital economy and related news disclosures will affect the timely decision-making of investors, especially in the field of financial science and technology, and then affect the efficiency and performance of listed companies [15]. Furthermore, the development of the digital economy and ICT can boost economic growth through technical progress in the CEE and EU-15 countries [16].

The second focuses on the evolution characteristics analysis of the industrial location. In essence, it is to study the geographical agglomeration of industry and the spatial transfer of development advantages. Studies have shown that the degree of industrial agglomeration is polarized, and the coastal areas are closer to the international market, which is more conducive to industrial agglomeration. The growth rate of manufacturing agglomeration in the central and western regions is higher, especially in the central region. The manufacturing industry (especially labor-intensive industries) has a trend of transferring to the central and western regions, showing a "flying goose" transfer mode, especially after 2004 [17]. In addition, some scholars have made a detailed discussion on the evolution characteristics of manufacturing agglomeration in a certain region. Since the western development strategy, the manufacturing industry in the western region has shown a "U-shaped" evolution trend, and the agglomeration degree has been improved, especially in the high-tech manufacturing industry [18]. Based on the perspective of evolution, Huang and Sun found that the agglomeration degree of manufacturing industry in the Beijing-Tianjin-Hebei region declined gradually from 2004 to 2013 [19].

The third focuses on the driving factor analysis of industrial location evolution. Existing studies have shown that: (1) Factors such as innate advantages, labor mobility costs, transportation costs, housing prices and institutions cause the dynamic changes of industrial agglomeration and dispersion. According to the traditional economic geography theory, industrial agglomeration is generally formed in areas with higher "innate comparative advantage" and better natural conditions such as minerals, transportation and climate [20]. Marshall mainly explored agglomeration from the perspective of "externality", believing that the externality factors generated by the input-output correlation, labor pool and knowledge spillover among enterprises are the important sources of industrial agglomeration [21]. However, the new trade theory and the new economic geography theory break through the limitations, which point out that the interaction of market potential, economies of scale and transportation costs promotes industrial agglomeration [22]. Labor prices and land costs will drive low-tech or labor-intensive enterprises to relocate [23]. Subsequently, institutional changes have gradually evolved into an important perspective, and govern-

ment actions represented by fiscal and taxation policies and industrial policies have also played an important role in manufacturing agglomeration [24]. (2) Technological factors can reshape the industrial geographic pattern. Sun and Hou constructed a new economic geography model to retest the traditional "geese theory" and found that AI changed the inter-industry transfer mode. In the meantime, the importance of labor costs on industrial layout was greatly weakened, and intelligent technology instead became an important force to build a new industrial pattern [25]. Likewise, the Internet has become a source of regional competitive advantage, which can increase consumer income to absorb labor force so as to attract enterprises to locate here, which is gradually becoming one of the important forces to reshape the economic geographic pattern of the manufacturing industry [26]. Moreover, ICT promotes industrial agglomeration, especially in high-tech industries with advanced ICT, after controlling other determinants of industrial agglomeration [27]. Contrary to the prevalent argument that ICT led to more agglomeration, Ioannides et al. developed a formal model and showed that the improvement of ICT will increase the dispersion of economic activities across cities, suggesting that city sizes will be more uniform [28]. In the digital era, the "transportation orientation" and "labor orientation" of industrial location are gradually transformed into "information orientation", and technology has gradually become one of the core factors of enterprise location choice [11].

To sum up, the existing literature reveals the effects of the digital economy, evolution characteristics and influencing factors of industrial agglomeration to a certain extent, but the impact of the digital economy on manufacturing agglomeration is less considered. Previous studies have paid more attention to the evolution of industrial location at the macro level, while only a small portion of the literature studies the impact of emerging technologies on industrial agglomeration from the micro perspective, and the internal mechanism is not clear. Based on this, this paper theoretically analyzes the internal mechanism of the impact of the digital economy on manufacturing agglomeration, and selects the data of China's listed companies from 2001 to 2020 to conduct empirical analysis at the micro level to explore its internal transmission mechanism. The specific contributions of this paper are as follows: First, in terms of research perspectives, unlike previous research, this study incorporates the digital economy and manufacturing aggregation into the same research framework for an in-depth analysis that provides a new perspective for the study of the causes of industrial agglomeration with certain novelty and topicality. Second, in terms of research methods, this study constructs the manufacturing agglomeration index from the enterprise scale, which can avoid the aggregation bias of macro-statistical data to a certain extent, further improve the estimation effect, and make the results more scientific and reasonable. Third, in terms of research content, a mediating effect model is selected to clarify the internal mechanisms of the digital economy on manufacturing agglomeration from the perspectives of transaction costs, market potential and knowledge spillover. Furthermore, multiple heterogeneity analysis is carried out from the aspects of region, urban agglomeration, city scale and enterprise so as to provide reference for industrial policy formulation. At present, China is in a critical period of economic transformation and high-quality development. This study can provide some useful reference for the promotion of the digital economy, the optimization of manufacturing layout and the high-quality development of the economy.

## 3. Theoretical Analysis and Research Hypothesis

At present, the importance of traditional factors, such as resource endowment and labor cost, has been greatly weakened, and intellectualization and digitalization have become important forces affecting the industrial layout [11]. In the era of the digital economy, the cross-regional flow of information, technology and capital is more convenient, and the location flexibility of manufacturing enterprises is greatly improved, but it does not mean that distance is no longer important [29]. Firstly, networks of regionally clustered businesses and institutions can offer two broad opportunities: formal exchanges of knowledge through market relationships, where proximity allows the establishment of closer

ties, and the informal exchange of knowledge in social networks of individuals [30]. It has been proven that formal exchanges of knowledge through digital technology are still constrained by geographical distance and location, and neighboring enterprises are more closely linked [31]. More importantly, the informal exchange of knowledge existing in the R&D process is often solidified in the regional innovation environment, interpersonal network and social culture, which are difficult to transmit through the digital technology [32]. In order to obtain more innovation and technology spillover brought about by the digital economy, enterprises will actively migrate to the periphery of enterprises with a higher degree of digital economy and form agglomerations. Secondly, the digital economy has accelerated the speed of information exchange, making time competition more important and leading to immediate demand and just-in-time production, which requires different manufacturing support enterprises to be close to each other to speed up supply. In the above process, the areas with a higher digital economy have a stronger attraction and continue to attract supporting enterprises to settle in and gather.

**Hypothesis 1.** *Digital economy can significantly promote the geographical agglomeration of the manufacturing industry.*

The theory of new economic geography holds that the reduction of transaction costs, the improvement of market potential and the enhancement of knowledge spillover can promote industrial agglomeration. Firstly, transaction cost savings is one of the key factors in promoting industrial agglomeration [33], and the advantages of the digital economy in reducing transaction costs have generally been affirmed. On the one hand, data disclosure and sharing under the development of the digital economy improve the matching degree between supply and demand and effectively reduce the cost of information search. On the other hand, the high-speed dissemination of data information makes the price more transparent, which reduces the bargaining cost in the process of signing the contract. The reduction of transaction costs makes the price advantage more prominent and the competitive advantage is further improved, thus attracting enterprises to settle in [25]. At the same time, according to the transaction cost theory, cost reduction is an important way for enterprises to obtain innovative resources and technologies from the outside [34]. The reduction of transaction costs brought by the digital economy not only enables enterprises to have more funds and energy to invest in innovation activities, but also enables enterprises to grasp market information timelier and accelerate the allocation of innovation factors so as to improve production efficiency and attract more enterprises to settle and gather [8].

Secondly, the theory of new economic geography holds that places with higher market potential are more likely to attract enterprises so as to obtain agglomeration benefits. In terms of supply potential, the digital economy has become an important driving force for strengthening, supplementing and extending the chain, which can greatly improve the level of information sharing and production collaboration among enterprises and further enhance the supply potential. In terms of demand potential, the digital economy has greatly shortened the distance between supply and demand; thus, individual consumers can participate in product design and manufacturing at any time to become producers and consumers. Meanwhile, digital technology can help enterprises achieve intelligent manufacturing, service-oriented manufacturing and large-scale customization so as to meet personalized and differentiated needs at a lower cost. With the increase in market demand scale, the upstream and downstream enterprises in the industrial chain are further gathered to form a new industrial network. Under the cumulative effect of circulation, the consumption potential is further released, promoting more new enterprises to settle in and gather [35].

Finally, the digital economy can also promote manufacturing agglomeration by enhancing knowledge spillover. On the one hand, the digital economy strengthens communication and exchanges among enterprises, especially in R&D cooperation, and establishes broader and deeper links so as to accelerate further knowledge spillover among enterprises. On

the other hand, the digital economy broadens the scope and improves the efficiency of imitation learning. Enterprises with strong technological strengths can produce demonstration effects in R&D, management, marketing and after-sales, which can spread over a wider range and at a faster speed, thus accelerating knowledge spillover. Knowledge spillover can make neighboring enterprises obtain technology spillover to the greatest extent, which is conducive to the geographical agglomeration of industries [36]. In the era of the digital economy, although knowledge spillover goes beyond the administrative geographical boundary, it still shows an obvious distance attenuation effect, and the choice of enterprise location still has a strong tendency of geographical agglomeration [37,38]. Therefore, in order to acquire technology or innovation spillover more quickly and widely, the manufacturing industry will actively gather near areas with a higher degree of digital economy.

**Hypothesis 2.** *The digital economy promotes the geographical agglomeration of the manufacturing industry by reducing transaction costs, enhancing market potential and promoting knowledge spillover.*

China has a vast territory, different regional characteristics and huge economic discontinuity, so the effect of the digital economy on manufacturing agglomeration is also heterogeneous. Firstly, in terms of economic globalization, participation in international trade and foreign direct investment are the two most important aspects of manufacturing globalization [39]. Foreign direct investment can improve the degree of capital supply and effectively optimize the capital mismatch caused by credit distortion, as well as alleviate the financing constraints of enterprises. In addition, the entry of an experienced R&D management team will enhance the degree of digitalization, and the regions with the first-mover advantage of the digital economy will be the most significant, thus attracting enterprises to settle in and enhancing manufacturing agglomeration. Secondly, the power of localization. Government intervention is an important way to rebuild local comparative advantage and an important element of localization [40]. Active government intervention will build a good ecology conducive to the development of the digital economy, which can effectively alleviate monopoly, negative externalities and other issues so as to maintain fair competition and attract enterprises. On the contrary, negative government intervention (such as local protection) will hinder the flow of products and factors, which can reduce the enthusiasm of external enterprises to enter. Therefore, in the process of the digital economy promoting manufacturing agglomeration, the role of localization forces cannot be ignored. Thirdly, high-quality human resources are the basis for accelerating the development of the digital economy and promoting innovation, as well as the key to whether enterprises can better rely on digital technology to achieve transformation and upgrading. Abundant and high-quality human capital can affect the breadth and depth of digital technology absorption and diffusion [8] and magnify the advantages of the digital economy to attract enterprises.

**Hypothesis 3.** *Globalization, localization and human capital play an important moderating role in the process of the digital economy, promoting the geographical agglomeration of the manufacturing industry.*

Based on the analysis of the above three aspects, this paper draws a theoretical mechanism map (Figure 2).

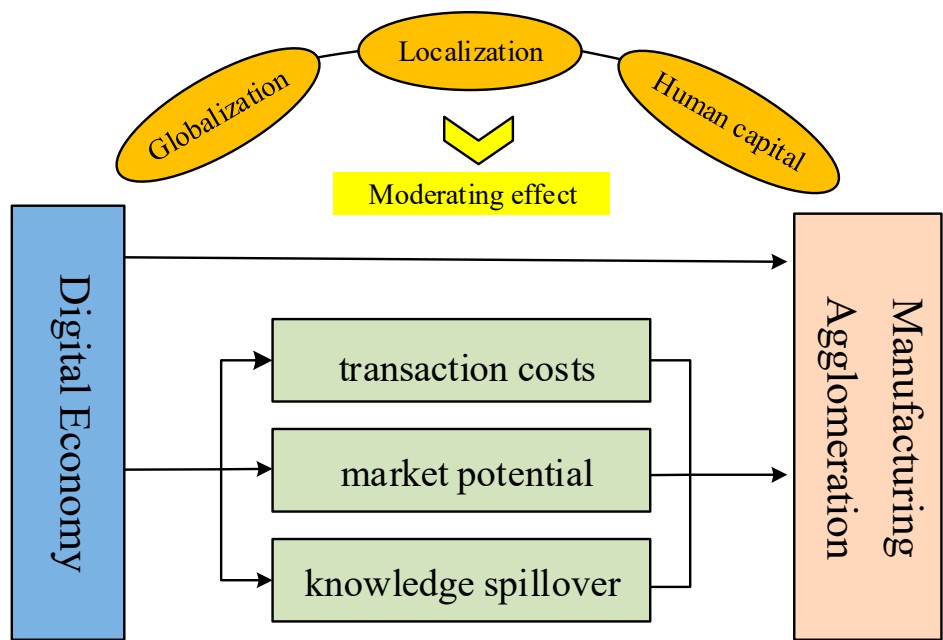

**Figure 2.** Theoretical mechanism of the digital economy promoting manufacturing agglomeration.

**4. Model, Variables and Data Resources**

*4.1. Model Construction*

The following benchmark regression model is constructed in this study to test the impact of the digital economy on manufacturing agglomeration:

$$lnagg\_m_{ijrt} = \alpha_0 + \alpha_1 lndig_{rt} + \alpha_2 lnX_{ijrt} + \mu_i + \delta_t + \varphi_t + \tau_t + \varepsilon_{it} \qquad (1)$$

In Equation (1), the subscripts *I, j, r, t*, respectively, represent enterprise, industry, city and year. $agg\_m_{ijrt}$ represents the agglomeration of the manufacturing industry. $dig_{rt}$ is the degree of digital economy of the city, and $X_{ijrt}$ are control variables. $\mu_i$, $\delta_t$, $\varphi_t$, $\tau_t$ respectively represent individual, time, industry and province fixed effects, so as to absorb as many fixed effects as possible, and $\varepsilon_{it}$ is the random error term.

In the aspect of mediation effect testing, this paper constructs the following mediation model based on relevant research [41], which can be expressed as follows:

$$mid_{ijrt} = \beta_0 + \beta_1 lndig_{ijrt} + \beta_2 lnX_t + \mu_i + \delta_t + \varepsilon_{it} \qquad (2)$$

$$lnagg\_m_{ijrt} = \gamma_0 + \gamma_1 lndig_{ijrt} + \gamma_2 mid_{ijrt} + \gamma_3 lnX_t + \mu_i + \delta_t + \varepsilon_{it} \qquad (3)$$

$mid_{ijrt}$ represents three mediating variables. In Equations (1)–(3), if the coefficient $\alpha_1$ is significant and both $\beta_1$ and $\gamma_2$ are significant, while satisfying $\gamma_1$ is smaller than $\alpha_1$, there is a partial mediating effect. If coefficient $\alpha_1$ is significant and both $\beta_1$ and $\gamma_2$ are significant, but $\gamma_1$ is not, there is a full mediating effect.

*4.2. Variables*

4.2.1. Explained Variables

The explained variable in this paper is the level of manufacturing agglomeration (*lnagg_m*). Location entropy is used to measure the spatial distribution of factors, reflecting the degree of industrial agglomeration in a specific region, and the larger the index, the higher the degree of industrial agglomeration. According to some classic studies, the locational entropy is generally measured from two perspectives: scale and quantity, and this paper measures it based on the enterprise scale perspective (total number of employees), which is calculated as the ratio of the number of employees in an industry (other than this

enterprise) in a local city to the proportion of manufacturing in the city to the proportion of employees in that industry nationwide [42]. The calculation formula is:

$$lnagg\_m_{ijrt} = \frac{\left(L_{jrt} - L_{ijrt}\right)/L_{rt}}{L_{jt}/L_t} \tag{4}$$

$L_{jrt}$ is the total number of employees of industry $j$ in region $r$ and period $t$. $L_{ijrt}$ is the total number of employees in enterprise $i$ of industry $j$ in region $r$ and period $t$. $L_{rt}$ is the total number of employees in the manufacturing industry in region $r$ and period $t$ and $L_{jt}$ is the total number of employees in industry $j$ in period $t$.

### 4.2.2. Explanatory Variables

The explanatory variable in this paper is the level of the digital economy (*lndig*). At present, it is difficult to accurately define the connotation of the digital economy, which is understood by many developed countries as the sum of economic activities based on modern information technology such as the Internet [43]. According to the relevant research of the China Academy of Information and Communications, it is believed that the digital economy is essentially the product of the deep integration of traditional economies and the Internet, and is the advanced form of the information economy [44]. The rapid development of the digital economy depends on the improvement of digital infrastructure, the expansion of digital scale and the promotion of digital applications [45]. Based on this, this paper takes the Internet as the main body [46], and constructs the evaluation index system of the digital economy from three dimensions: digital infrastructure, digital development scale and digital application degree (Table 1). On this basis, principal component analysis is used to reduce the dimensionality, so as to calculate the comprehensive index of the digital economy.

**Table 1.** Evaluation index system of digital economy.

| Level I Indicators | Secondary Indicators | Unit |
|---|---|---|
| digital infrastructure | Length of long-distance optical cable line per square kilometer | km/km$^2$ |
|  | Internet broadband access port per 100 people | PCs |
|  | Mobile telephone exchange capacity per household | % |
| digital development scale | Express business volume per capita | PCs |
|  | Total amount of telecommunication business per capita | Yuan |
| digital application degree | proportion of mobile phone users | % |
|  | Internet penetration rate | % |
|  | Proportion of personnel in information transmission, software and information technology services | % |

### 4.2.3. Mediating Variables

(1) Transaction cost (*lncost*): this paper uses asset specificity to reflect transaction costs. Firms with higher asset specificity face higher "hold-up" risk and a higher probability of being "ripped off" by counterparties, thus facing higher external transaction costs. Referring to the research of Collis and Montgomery, this paper uses the ratio of intangible assets to total assets to measure enterprise asset specificity [47]. The data on intangible assets come from the WIND database, which refers to the identifiable non-monetary assets owned or controlled by enterprises without physical objects. (2) Market potential (*lnmp*). In view of the availability of data, this paper draws on Harris' study to measure the market potential from the demand side and holds that the market potential is to measure the scale of market demand faced by cities or the associated benefits of inter-city markets and is the weighted sum of purchasing power [48]. Specifically, the weight is the reciprocal of the distance, and it is directly proportional to the GDP of the region and its neighbors and

inversely proportional to the distance from other regions to the region. The calculation formula is as follows:

$$mp_{it} = \sum_{j=1}^{n} \frac{GDP_{jt}}{d_{ij}} + \frac{GDP_{it}}{d_{ii}} \tag{5}$$

$$d_{ii} = \frac{2}{3}\sqrt{\frac{area_i}{\pi}} \tag{6}$$

$mp_{it}$ denotes the market potential of city $i$ in period $t$. $GDP_{jt}$ and $GDP_{it}$ represents the GDP of cities $j$ and $i$ ($i \neq j$), $d_{ij}$ is the straight-line distance (road distance) between city $j$ and $i$, and $area_i$ is the land area of city $i$. $d_{ii}$ is the diameter of city $i$. (3) Knowledge spillover. The theoretical meaning of knowledge spillover is relatively clear, but the empirical measurement faces greater difficulties, and the academic community has not yet formed a unified measurement standard. This paper refers to the classical literature, using the number of patent citations to measure the knowledge spillover effect (*lnkno*) [49], which can capture the "literature trail" and observe the knowledge spillover process.

### 4.2.4. Control Variables

Control variables. In addition to the core explanatory variables, the following control variables are also used in this study. Enterprise level: Enterprise age (*lnage*): the year in which the enterprise is located minus the year of establishment plus one; Asset-liability ratio (*lnlev*): the ratio of total liabilities to total assets; Current ratio (*lnliq*): the ratio of current assets to current liabilities; Cash ratio (*lncash*): the ratio of net cash flow from operating activities to total assets. City level: Economic development level (*lnpgdp*): GDP per capita; Financial potential (*lnfin*): which is measured by the balance of deposits and loans of financial institutions at the end of the year; Market development level (*lnmark*), calculated based on Wang and Fan's marketization indicators [50]; Foreign direct investment (*lnfdi*): the ratio of actual foreign direct investment to GDP of each prefecture-level city in the current year; Government intervention (*lngov*), which is measured by the proportion of local government general budget expenditure to GDP. Human capital (*lnhum*), measured by the proportion of ordinary undergraduates and above (%).

### 4.3. Data Resources

This paper uses the data of A-share listed companies in China's manufacturing industry from 2001 to 2020 as the initial research sample and excludes the samples of ST, PT and insolvency. The company financial data are obtained from the CSMAR and Wind databases, and the regional-level data have mainly come from the Statistical Yearbook of Chinese Cities over the years. For a small number of missing values, the polynomial fitting method is used to interpolate and fill. In order to eliminate the influence of heteroscedasticity, the data were processed logarithmically. The descriptive statistics were performed on the variables using Stata 16. Table 2 shows the statistics of each variable, and all variables are relatively smooth.

**Table 2.** Descriptive statistics.

| Variables | Number of Samples | Mean | Std. Dev. | Min | Max |
|---|---|---|---|---|---|
| lnagg_m | 24,615 | 0.310 | 0.590 | −1.507 | 0.934 |
| lndig | 24,615 | 0.026 | 0.057 | 0.000 | 0.328 |
| lncost | 24,605 | 4.543 | 0.164 | 4.037 | 5.297 |
| lnmp | 20,889 | 7.548 | 1.298 | 4.339 | 9.668 |
| lnkno | 15,751 | 5.399 | 2.048 | −1.061 | 9.409 |
| lnage | 24,599 | 2.744 | 0.419 | 1.099 | 3.526 |
| lnlev | 24,613 | 1.040 | 0.606 | 0.057 | 2.829 |
| lnliq | 24,613 | 0.615 | 0.740 | −0.947 | 2.726 |
| lncash | 24,609 | −0.465 | 1.081 | −3.283 | 2.334 |

**Table 2.** *Cont.*

| Variables | Number of Samples | Mean | Std. Dev. | Min | Max |
|---|---|---|---|---|---|
| lnpgdp | 21,011 | 11.070 | 0.752 | 8.977 | 12.150 |
| lnfin | 19,748 | 18.910 | 1.534 | 15.560 | 21.590 |
| lnhum | 18,618 | 0.779 | 0.886 | −1.531 | 2.334 |
| lnmark | 21,188 | 2.367 | 0.319 | 1.386 | 2.890 |
| lnfdi | 20,590 | −5.822 | 0.957 | −9.083 | −4.360 |
| lngov | 18,095 | 7.209 | 0.389 | 5.746 | 10.060 |

## 5. Results and Discussion

### 5.1. Baseline Regression Analysis

Table 3 reports the benchmark regression results of the impact of the digital economy on manufacturing agglomeration, with columns (1) controlling for fixed effects only, columns (2) and (3) adding enterprise-level and city-level control variables, respectively. The results show that the coefficients of the digital economy are all significantly positive at the level of 1%. Taking the results in column (3) as an example, when the level of the digital economy increases by 1%, the degree of manufacturing agglomeration increases by 0.1162%, which verifies hypothesis 1.

**Table 3.** Baseline regression results.

| Variables | (1) | (2) | (3) |
|---|---|---|---|
| | Lnagg_m | Lnagg_m | Lnagg_m |
| lndig | 0.1351 *** | 0.1332 *** | 0.1162 *** |
| | (0.0460) | (0.0459) | (0.0386) |
| lnage | | −0.0794 ** | −0.1524 *** |
| | | (0.0366) | (0.0398) |
| lnlev | | 0.0016 | −0.0031 |
| | | (0.0108) | (0.0108) |
| lnliq | | −0.0152 | −0.0177 |
| | | (0.0122) | (0.0129) |
| lncash | | 0.0010 | 0.0048 |
| | | (0.0057) | (0.0062) |
| lnpgdp | | | −0.0932 *** |
| | | | (0.0331) |
| lnfin | | | −0.1296 *** |
| | | | (0.0268) |
| lnhum | | | −0.1152 *** |
| | | | (0.0179) |
| lnmark | | | 0.0417 |
| | | | (0.0794) |
| lnfdi | | | −0.0098 * |
| | | | (0.0059) |
| lngov | | | 0.0089 |
| | | | (0.0117) |
| _cons | 0.3068 *** | 0.5325 *** | 4.0441 *** |
| | (0.0012) | (0.1007) | (0.5573) |
| individual, time, industry and province | Control | Control | Control |
| Number of samples | 24,264 | 24,242 | 15,978 |
| $\overline{R^2}$ | 0.9322 | 0.9324 | 0.9536 |

Note:*, **, *** are significant at 10%, 5% and 1% levels, respectively. Same after table.

### 5.2. Robustness Test and Endogenous Treatment

In order to ensure the robustness of the above research results, this paper conducts tests from the following aspects, and the results are shown in Table 4.

**Table 4.** Robustness Test.

| Variables | (1) | (2) | (3) | (4) |
|---|---|---|---|---|
| | Lnagg_m2 | Lnagg_m2 | Lnagg_m | Lnagg_m |
| lndig | 0.0790 ** | 0.0448 * | | 0.1453 *** |
| | (0.0320) | (0.0235) | | (0.0455) |
| lndig2 | | | 0.0199 ** | |
| | | | (0.0095) | |
| _cons | 0.0753 *** | 2.5074 *** | 2.2814 *** | 2.1974 *** |
| | (0.0008) | (0.3690) | (0.5290) | (0.5973) |
| Controls | Uncontrol | Control | Control | Control |
| individual, time, industry and province | Control | Control | Control | Control |
| Number of samples | 24,264 | 15,978 | 12,431 | 13,251 |
| $\overline{R^2}$ | 0.9140 | 0.9418 | 0.9702 | 0.9019 |

Note: *, **, *** are significant at 10%, 5% and 1% levels, respectively.

Substitution of the explained variable. It may be difficult to comprehensively measure the change in manufacturing agglomeration by constructing the location entropy only from the perspective of scale (number of employees). Based on this, this paper refers to Wang's study and uses the "number of enterprises" to construct the location entropy of the manufacturing industry (*lnagg_m2*) from the perspective of quantity [8]. Columns (1) and (2) show the results before and after adding the control variables. It can be seen that the digital economy has significantly promoted manufacturing agglomeration, both in terms of scale and quantity.

Substitution of explanatory variables. Based on Zhao's research, the index system of the digital economy is constructed from five aspects: telecommunication business income, information transmission computer services, Internet broadband access users, mobile phone users and an inclusive financial index, and the comprehensive level of the digital economy (*lndig2*) is calculated by the entropy method [51]. The results are shown in column (3), which is consistent with the conclusion based on the benchmark results.

Delete special samples. Whether in terms of economic development, technology, or openness to the world, the development of municipalities directly under the Central Government is superior to that of general prefecture-level cities, so the special samples are removed. The regression results are shown in column (4), and the results are still robust.

The empirical results above may have endogeneity problems due to reverse causality, and in order to solve this problem this paper uses two instrumental variable methods. Firstly, based on the study of Guo and Luo, this paper selects the explanatory variables with one period lag as the first instrumental variable [52], and the test results are shown in columns (1) and (2) of Table 5, where the Kleibergen Paap rk LM statistic is significant at the 1% level, rejecting the unidentifiable original hypothesis. At the same time, the Cragg–Donald Wald F statistic is greater than the critical value of the Stock-Yogo weak instrumental variable identification F test at the 10% significance level (16.38), rejecting the original hypothesis of weak instrumental variables. Therefore, the instrumental variable is reasonable and reliable, and the results are consistent with the benchmark regression results. Secondly, this paper uses the Bartik method to construct instrumental variables with reference to some classical studies [53]. Specifically, the number of fixed telephones per 10,000 people in each city in 1984 is taken as the base period share, and the growth rate of national Internet users is set as the exogenous weight to reweight the level of digital economy in the region. The formula is as follows: $IV_{pt} = \ln(w_p^{1984}(1 + Grate_t))$, where p and t denote city and year. Moreover, $w_p^{1984}$ is the number of fixed telephones per 10,000 people in each city in 1984, and Grate indicates the growth rate of national Internet users relative to the base period. The Bartik instrument will not be correlated with other residual terms and can better solve the endogeneity problem. The results are shown in columns (3) and (4), which pass the unidentifiable test and the weak instrumental variable test. In summary, the estimation results of the above two methods show that the digital

economy significantly promotes manufacturing agglomeration; that is, the results of this paper are still robust and reliable.

**Table 5.** Endogenous Treatment.

| Variables | IV (One Period Lag) | | IV (Bartik) | |
|---|---|---|---|---|
| | (1) | (2) | (3) | (4) |
| First stage IV | 0.4367 *** | | 0.1765 *** | |
| | (0.0077) | | (0.0088) | |
| lndig | | 0.2761 *** | | 0.7239 *** |
| | | (0.0640) | | (0.1696) |
| Controls | Control | Control | Control | Control |
| individual, time, industry and province | Control | Control | Control | Control |
| Kleibergen-Paap rk LM statistic | | 2931.461 *** | | 49.976 *** |
| Cragg-Donald Wald F statistic | | 3228.07 | | 456.274 |
| Number of samples | 14,608 | 14,608 | 14,879 | 14,879 |
| $\overline{R}^2$ | | 0.0527 | | 0.0279 |

Note: *** is significant at 1% levels.

*5.3. Heterogeneity Analysis*

5.3.1. Enterprise Heterogeneity

Based on the micro-characteristics of enterprises, this paper explores the heterogeneity from two aspects: enterprise scale and factor intensity, and the results are shown in Table 6. According to the classification of enterprise scale in Wind database, they are divided into large, small and medium-sized enterprises. Columns (1) and (2) show that the digital economy significantly promotes the agglomeration of large manufacturing enterprises, while the impact on small and medium-sized manufacturing enterprises is positive but not significant. Large enterprises have unparalleled advantages in resource endowment, innovation and R&D and industry chain synergy and are more likely to generate economies of scale and scope. Studies have also proved that "Internet+" can promote the employment growth of large enterprises, but it has a suppressive effect on small and micro enterprises [54]. In addition, it can also be explained based on Schumpeter's innovation theory, that is, in the era of the digital economy, driven by the advantages of increasing returns to scale and high-risk tolerance, the innovation motivation of large enterprises is stronger and the Matthew effect is more prominent. Thus, the agglomeration effect of the digital economy on the manufacturing industry is more easily reflected in large enterprises. In terms of factor intensity, according to the Classification of High-tech Industries (Manufacturing) (2017), enterprises are divided into high-tech and low-tech manufacturing. Columns (3) and (4) show that the digital economy can significantly promote the agglomeration of the high-tech manufacturing industry. On the one hand, the selection effect of the digital economy will attract high-tech and high-productivity enterprises to enter and make low-productivity or old-technology enterprises withdraw from the market or locate in the peripheral areas. On the other hand, high-tech manufacturing industries are more dependent on digital and intelligent technology, for it can effectively promote continuous innovation of products and technologies of enterprises [55]. To some extent, digital economy technology can also be regarded as a part of corporate R&D innovation. Therefore, the higher the level of the digital economy, the more it can promote the agglomeration of the high-tech manufacturing industry.

5.3.2. Regional Heterogeneity

Industrial agglomeration always occurs in a specific space; thus, manufacturing agglomeration is necessarily affected by local characteristics. Columns (1) and (2) of Table 7 show that the promotion effect of the digital economy on manufacturing agglomeration in central and western China is significantly positive at the 1% level. The western region has obvious advantages in land and labor force, but the digital economic foundation is

weak and lacks a complete industrial chain. In order to save on transportation costs and improve risk resistance, affiliated enterprises tend to be close to each other in space to form agglomeration advantages.

**Table 6.** Heterogeneity analysis: enterprise heterogeneity.

| Variables | (1) | (2) | (3) | (4) |
|---|---|---|---|---|
| | Large-Sized | Small and Medium-Sized | High-Tech | Low-Tech |
| lndig | 0.1175 *** | 0.0889 | 0.0926 ** | 0.1061 |
| | (0.0389) | (0.1093) | (0.0438) | (0.0699) |
| _cons | 4.1102 *** | 3.3794 *** | 2.9915 *** | 5.6657 *** |
| | (0.6356) | (1.0921) | (0.5998) | (1.0325) |
| Controls | Control | Control | Control | Control |
| individual, time, industry and province | Control | Control | Control | Control |
| Number of samples | 12,984 | 2994 | 10,008 | 5886 |
| $\overline{R^2}$ | 0.9507 | 0.9652 | 0.9634 | 0.9341 |

Note: **, *** are significant at 5% and 1% levels, respectively.

**Table 7.** Heterogeneity analysis: regional heterogeneity.

| Variables | (1) | (2) | (3) | (4) | (5) | (6) |
|---|---|---|---|---|---|---|
| | East | Central and Western | Hu Huanyong Line East | Hu Huanyong Line West | Large Cities | Small and Medium Cities |
| lndig | 0.0056 | 0.4217 *** | 0.1063 *** | 0.4085 * | 0.0041 | 0.1583 *** |
| | (0.0316) | (0.1200) | (0.0391) | (0.2149) | (0.0472) | (0.0564) |
| _cons | 2.5676 *** | 3.2528 *** | 3.9269 *** | 1.6970 | 5.0245 *** | 1.2156 |
| | (0.5588) | (1.0480) | (0.6388) | (1.6191) | (0.6180) | (0.8298) |
| Controls | Control | Control | Control | Control | Control | Control |
| individual, time, industry and province | Control | Control | Control | Control | Control | Control |
| Number of samples | 10,642 | 5336 | 15,351 | 627 | 7861 | 8117 |
| $\overline{R^2}$ | 0.9755 | 0.8499 | 0.9549 | 0.7961 | 0.9726 | 0.8830 |

Note: *, *** are significant at 10% and 1% levels, respectively.

At present, the unbalanced development of China's manufacturing industry is relatively serious, especially the unbalanced distribution on both sides of the "Hu Huanyong Line", forming an obvious industrial gradient with faster development in the southeast and slower development in the northwest [56]. Columns (3) and (4) show that the digital economy has a significant positive impact on both sides of the Hu Huanyong Line. It is more conducive to promoting the manufacturing agglomeration on the west side, which can help the manufacturing industry break through the Line to a certain extent and realize the balanced development.

In addition, according to the Notice of the State Council on Adjusting the Standards for Dividing the Size of Cities in 2014, those with more than 3 million permanent residents are defined as large cities, others are small and medium-sized cities. Columns (5) and (6) show that the digital economy plays a significantly positive role in small and medium-sized cities. Although big cities are more inclusive, competitive and convenient, with the expansion of agglomeration effects, housing prices, land prices, pollution and other congestion effects that are not conducive to the further agglomeration of enterprises. By contrast, small and medium-sized cities have more potential, which is an important carrier to undertake industrial transfer and develop the real economy. It is also an important force in promoting high-quality economic and social development.

### 5.4. Mediating Effect Analysis

The above theoretical analysis shows that the digital economy promotes manufacturing agglomeration by reducing transaction costs, improving market potential and promoting knowledge spillover. The empirical regression results are shown in Table 8.

**Table 8.** Mediating effect results.

| Variables | (1) Lncost | (2) Lnagg_m | (3) Lnmp | (4) Lnagg_m | (5) Lnkno | (6) Lnagg_m |
|---|---|---|---|---|---|---|
| lndig | −0.1314 * | 0.0762 ** | 0.0249 ** | 0.1126 *** | 0.8581 *** | 0.1138 *** |
| | (0.0762) | (0.0311) | (0.0109) | (0.0269) | (0.1576) | (0.0394) |
| lncost | | −0.0173 *** | | | | |
| | | (0.0037) | | | | |
| lnmp | | | | 0.1400*** | | |
| | | | | (0.0207) | | |
| lnkno | | | | | | 0.0028 * |
| | | | | | | (0.0038) |
| _cons | −0.6970 | 4.2781 *** | −3.8358 *** | 4.5925 *** | −10.2578 *** | 3.6834 *** |
| | (0.6233) | (0.2546) | (0.0969) | (0.2516) | (1.7318) | (0.2589) |
| Controls | Control | Control | Control | Control | Control | Control |
| individual, time, industry and province | Control | Control | Control | Control | Control | Control |
| Number of samples | 13,893 | 13,893 | 15,916 | 15,916 | 10,422 | 13,722 |
| $\overline{R^2}$ | 0.7628 | 0.9479 | 0.9980 | 0.9539 | 0.8360 | 0.9548 |

Note: *, **, *** are significant at 10%, 5% and 1% levels, respectively.

Specifically, Columns (1) and (2) are the results of transaction cost as the mediating variable, in which the coefficient of lndig is significantly negative, indicating that the digital economy can reduce the transaction cost. The regression coefficient of lncost in column (2) is significantly negative, and the coefficient of lndig is lower than the benchmark regression, indicating that the digital economy can promote manufacturing agglomeration by reducing transaction costs. Columns (3) and (4) are the regression results with market potential as the mediating variable. It is easy to see that the regression coefficient of lndig is significantly positive, indicating that digital economy has a positive impact on the promotion of market potential, and the coefficient of market potential on manufacturing agglomeration in column (4) is also significantly positive. It shows that the digital economy can indirectly promote manufacturing agglomeration by improving market potential. Similarly, it can also be inferred from columns (5) and (6) that the digital economy promotes manufacturing agglomeration through the positive mediating effect of knowledge spillover.

### 5.5. Moderating Effect Analysis

Through the above analysis, it can be seen that there are many aspects of heterogeneity in the process of the digital economy promoting manufacturing agglomeration. What causes such heterogeneity? This requires the exploration of the differences in local environments. Firstly, it is globalization. According to the above theoretical analysis, this paper focuses on economic globalization, which is dominated by the exchange of goods and services and the inflow of foreign capital, so it is measured by foreign direct investment (*lnfdi*) [7] (Table 9). Column (1) shows that the multiplier coefficient of the digital economy and globalization is positive and significant, indicating that the higher the degree of globalization, the stronger the effect of the digital economy on promoting manufacturing agglomeration. Foreign direct investment can not only bring capital and employment directly but also bring advanced management concepts and production technology to enterprises, attracting upstream and downstream enterprises to enter. Secondly, in terms of localization, according to the theoretical analysis above, this paper selects government intervention (*lngov*) to measure the degree of localization [39]. Most studies have proven that the decentralization of economic decision-making power and the evaluation of local government officials based on economic performance will enhance the local protection-

ism of the government, resulting in the obstruction of factor flow and the decline of the manufacturing agglomeration. Column (2) shows that the multiplier coefficient of the digital economy and localization is significantly positive, indicating that the higher the degree of localization, the stronger the agglomeration effect of the digital economy on the manufacturing industry, which is different from some previous studies. The possible reason is that, at the present stage, government subsidies and preferential policies in digitalization and investment attraction promote manufacturing agglomeration to a certain extent. Finally, the digital economy cannot be separated from the support of talents, especially the participation of digital talents, which is measured by human capital(*lnhum*). Column (3) shows that the multiplier coefficient of digital economy and human capital is significantly positive, indicating that the leverage of human capital will be highly valued.

**Table 9.** Moderating effects of globalization, localization and human capital.

| Variables | (1) Lnagg_m | (2) Lnagg_m | (3) Lnagg_m |
|---|---|---|---|
| lndig | 0.5615 ** | −0.8748 | 0.0386 |
| | (0.2463) | (0.6314) | (0.0364) |
| lnfdi | −0.0229 | −0.0100 * | −0.0100 * |
| | (0.0121) | (0.0059) | (0.0059) |
| lngov | 0.0091 | 0.0114 | 0.0121 |
| | (0.0117) | (0.0239) | (0.0117) |
| lnhum | −0.1142 *** | −0.1143 *** | −0.2312 |
| | (0.0179) | (0.0181) | (0.0360) |
| c.lndig#c.lnfdi | 0.0790 * | | |
| | (0.0425) | | |
| c.lndig#c.lngov | | 0.1355 * | |
| | | (0.0846) | |
| c.lndig#c.lnhum | | | 0.0917 ** |
| | | | (0.0467) |
| _cons | 4.0041 *** | 4.0992 *** | 4.0089 *** |
| | (0.5603) | (0.5624) | (0.5580) |
| Controls | Control | Control | Control |
| individual, time, industry and province | Control | Control | Control |
| Number of samples | 15,978 | 15,978 | 15,978 |
| $\overline{R^2}$ | 0.9537 | 0.9537 | 0.9537 |

Note: *, **, *** are significant at 10%, 5% and 1% levels, respectively.

### 5.6. Futher Discussion

According to Section 5.1, the digital economy has had a significant positive direct effect on manufacturing agglomeration during the sample period; that is, the digital economy has gradually become an important force in promoting the evolution of industrial layout. First of all, according to Section 5.3.1, the digital economy has significantly promoted the agglomeration of high-tech manufacturing, which is consistent with the works of Wang et al., Hong and Fu, and Wei et al. [8,27,57]. Wang used the micro-enterprise data of Tian Eye Check, combined with web crawler technology and text capture methods, to find that industrial intelligence has a selective bias to the distribution of manufacturing enterprises, which can encourage AI enterprises to gather [8]. Hong and Fu also pointed out that high-tech industries with advanced information and communication technologies also tend to gather [27]. The possible explanation is that, on the one hand, compared with low-tech manufacturing industries, high-tech manufacturing industries need timelier and more extensive R&D innovation, and Audretsch and Feldman pointed that industries that emphasize research and development (R&D) are more likely to concentrate in an area [58]. On the other hand, workers in high-tech industries need more face-to-face contacts with their peers to exchange ideas and thus benefit more from information spillovers than workers in low-tech industries [59]. Based on this, when implementing local policies (such as industrial park policies), it is necessary to consider the differences in the agglomeration

effect of the digital economy on enterprises in different industries so as to enhance the pertinence of policies. In addition, according to Section 5.3.2, the digital economy has significantly promoted the agglomeration of manufacturing industry in the central and western regions, small and medium-sized cities and the west side of the Hu Huanyong line, which can narrow the regional differences in manufacturing development to a certain extent and then promote the coordinated development of regions. This research result enriches and expands the research of An and Yang [26] but is contrary to the research results of Wu et al. [60]. The possible explanation is that, compared with big cities and eastern regions, the dividends of the digital economy in small and medium-sized cities and central and western regions have not been fully released. Especially, the eastern regions have had the lion's share of the digital economy to date, but growth rates are fastest in the central and western regions, so there is more room for improvement and the agglomeration effect is relatively stronger. In a word, the digital economy provides an opportunity for the coordinated development of regional economies in China.

## 6. Conclusions and Recommendations

### 6.1. Conclusions

Under the background of "re-industrialization", the world has entered the era of manufacturing competition, while at the same time, the digital economy has a profound impact on industrial location (manufacturing agglomeration), which brings new opportunities for reshaping manufacturing competitiveness and high-quality economic development. Based on the data of manufacturing enterprises from 2001 to 2020 in China, this paper analyzes the effect and internal mechanisms of the digital economy on manufacturing agglomeration from both theoretical and empirical perspectives.

The research conclusions are as follows: First, the digital economy has significantly promoted China's manufacturing agglomeration, which has obvious heterogeneity and plays a more significant role in the samples of large and high-tech manufacturing enterprises. Meanwhile, the agglomeration effect is more significant in the samples of small and medium-sized cities, the central and western regions, and the west side of the "Hu Huanyong Line", which can help the layout of the manufacturing industry break through the "Hu Huanyong Line" to a certain extent and achieve balanced development. Second, the mechanism test shows that the digital economy promotes manufacturing agglomeration mainly by reducing transaction costs, improving market potential and enhancing knowledge spillover. Third, globalization, localization and the level of human capital have a significant positive moderating effect on the process of the digital economy promoting manufacturing agglomeration. This paper reveals the effect and internal mechanism of the digital economy on manufacturing agglomeration, and the conclusions have important implications for the formulation of digital policy and regional industrial policy, which can further promote the process of high-quality development of the manufacturing industry and regional coordinated development.

### 6.2. Recommendations

Based on the above empirical results, this study proposes the following policy recommendations.

Firstly, the government should continue to accelerate the development process of the regional digital economy and give full play to its agglomeration effect. This paper finds that the agglomeration effect of the digital economy is more significant in the central and western regions and in small and medium-sized cities. Compared with the eastern regions and big cities, the digital infrastructure and digital technology capability required by the development of digital economy in these areas are still insufficient. Therefore, the government should increase investment in new digital infrastructure such as 5G, big data and artificial intelligence and moderately advance the construction of digital infrastructure so as to better promote the coordinated development of regions.

Secondly, this paper shows that the digital economy can significantly promote the agglomeration of high-tech manufacturing industries, which is the key to deep industrialization and high-quality development of the manufacturing industry, so it is necessary to build a high-tech manufacturing cluster with the help of the digital economy. Specifically, the government should build a platform for sharing knowledge resources and services in industrial clusters with the help of digital technology to promote the interconnection of high-tech manufacturing industries and realize the sharing of digital resources so as to maximize the agglomeration effect of the digital economy and promote the high-quality development of the manufacturing industry.

Thirdly, it should continue to promote the development of economic globalization and localization and better play the positive role of market potential and knowledge spillover in manufacturing agglomeration. For the central and western regions, a number of regional and cross-regional digital infrastructure projects should be planned and constructed as a whole so as to further expand the market potential with the strength of the digital economy and promote the agglomeration of the manufacturing industry. For the eastern region, it is necessary to conform to the trend of globalization and firmly seize the opportunities brought about by the new round of scientific and technological revolution. Local governments should take advantage of digital technology to actively build an integrated economic network with the central and western regions, strengthening regional cooperation in scientific and technological innovation, so as to promote knowledge spillover and better promote regional coordinated development.

*6.3. Limitations and Future Research*

Due to the limitation of data availability, this paper only selected indexes regarding the aspects of digital infrastructure, digital development scale and digital application degree to construct the evaluation index system of the digital economy. However, digital economy is a relatively broad concept, and as human society gradually enters a new stage marked by digitalization, the connotation of digital economy continues to expand and extend. The optimization of the evaluation index and consideration of other related factors are important issues that need to be solved in further research.

Furthermore, the digital economy promotes virtual agglomeration, especially in producer services, which are highly related to the manufacturing industry, but the internal mechanism is not clear, so it is urgent to further deepen the research on the internal mechanism in the future. In addition, there may be differences in the heterogeneity generated by different external environments, especially in different cyclical backgrounds [61], thus the heterogeneity analysis should be discussed and explained in a more comprehensive and detailed way in the future study.

**Author Contributions:** Conceptualization, M.W. and M.Z.; methodology, M.W. and M.Z.; software, H.C.; validation, M.W. and M.Z. and H.C.; formal analysis, D.Y.; writing—original draft preparation, M.W. and M.Z.; writing—review and editing, H.C. and D.Y.; visualization, M.W.; funding acquisition, D.Y. All authors have read and agreed to the published version of the manuscript.

**Funding:** This research was funded by National Natural Science Foundation of China (No. 71973083), National Natural Science Foundation of China (No. 42101240), Humanities and Social Science Research and Planning Foundation of the Ministry of Education (No. 19YJA790109).

**Institutional Review Board Statement:** Not applicable.

**Informed Consent Statement:** Not applicable.

**Data Availability Statement:** The sample data are sourced from the corresponding years of the "China Statistical Yearbook" and "Statistical Yearbook of Chinese Cities", and the company financial data are obtained from the CSMAR and Wind database. All data can be obtained by email from the corresponding author.

**Conflicts of Interest:** The authors declare no conflict of interest.

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
