# Peer review of "How Does Digital Economy Promote the Geographical Agglomeration of Manufacturing Industry?"

_sustainability, doi:10.3390/su15021727_

Round 1

Reviewer 1 Report

It is of great theoretical and practical significance to study the impact of digital economy on the geographical agglomeration of manufacturing industry. Taking the listed manufacturing companies in China as samples, this paper analyzes the impact of digital economy on the geographical agglomeration of manufacturing industry from both theoretical and empirical perspectives, and makes in-depth research and analysis from the aspects of impact, transmission mechanism, heterogeneity, etc. The logical level of the paper is reasonable, but it should be improved as follows:

1. The introduction should supplement the relevant literature.

Although this paper elaborated the related literature, this paper did not summarize the digital economy literature enough. The research of digital economy involves a series of problems including the measurement of digital economy. The following documents can be referred to:

Jiehua Ma, Zhenghui Li. Measuring China's urban digital economy[J]. National Accounting Review, 2022, 4(4): 329-361. doi: 10.3934/NAR.2022019

Yanting Xu, Tinghui Li. Measuring digital economy in China[J]. National Accounting Review, 2022, 4(3): 251-272. doi: 10.3934/NAR.2022015

Meanwhile, this paper involves listed companies, but digital finance and its related news have an impact on listed companies. These documents also provide a theoretical basis for this study. References:

Li, Z., Yang, C., Huang, Z. (2022). How does the fintech sector react to signals from central bank digital currencies? Finance Research Letters. https://doi.org/10.1016/j.frl.2022.103308

Li Z., Chen H. & Mo B. (2022), Can digital finance promote urban innovation? Evidence from China, Borsa Istanbul Review, https://doi.org/10.1016/j.bir.2022. 10.006.

2. Part 4 explains the empirical results in Table 3. This paper expounds the role of digital economy in listed companies, but the decision-making process of listed companies is strongly related to corporate R&D and other behaviors, especially in the process of financial asset allocation. Therefore, the role of R&D innovation should be considered in the process of conclusion interpretation, and digital economy technology can also be regarded as a part of corporate R&D innovation. Specific references:

Liu Y, Failler P, Ding Y (2022) Enterprise financialization and technological innovation: Mechanism and heterogeneity. PLoS ONE 17(12): e0275461. https://doi.org/10.1371/journal.pone.0275461

However, the geographical agglomeration of manufacturing industry is still related to the total factor productivity of enterprises. For details, please refer to the following literature:

Li, Z., Zou, F., & Mo, B. (2021). Does mandatory CSR disclosure affect enterprise total factor productivity?. Economic Research-Ekonomska Istraživanja, 1-20. doi:10.1080/1331677X. 2021.2019596

3. Part 4 discusses the intermediary effect and heterogeneity. The discussion of intermediary effect and heterogeneity is the result of different conditions, so there may be differences in the heterogeneity generated by different external environments, especially in different cyclical backgrounds. The author should supplement relevant explanations or explain the problem, and relevant literature is as follows:

Li, T., Li, X., & Liao, G. (2022). Business cycles and energy intensity. Evidence from emerging economies. Borsa Istanbul Review, 22(3), 560-570.. doi: 10.1016/j.bir.2021.07.005

Liu, Y., Li, Z., & Xu, M. (2020). The influential factors of financial cycle spillover: evidence from China. Emerging Markets Finance and Trade, 56(6), 1336-1350. doi: 10.1080/1540496x.2019.1658076 

4. It is suggested that the author should carefully sort out the full text.

Reviewer 2 Report

Research summary:

 In this research the authors have tried to establish a relation between geographical agglomeration of manufacturing industry and digital economy. As a whole, this is quite an interesting paper however it  has a few shortcomings. By overcoming mentioned inadequacies the authors can further enhance the quality of this research.

I have listed my major concerns below:

è Section 1 is too lengthy and does not provide a clear sense of the topic. Authors are directed to rewrite this section by building the arguments in a logical manner. Further, research questions also need to be categorically mentioned in this Section.

 è The novelties of the proposed research is not obvious compared to the existing studies. Therefore, authors are advised to provide a comparison table clearly mentioning the specific contributions of this research. It would be better if Section 1 is split into “Introduction” and “Literature Review” Sections.

 è Research question mentioned in the introduction section, must be quoted in the result and discussion section comprehensively. (very important). 

Reviewer 3 Report

The paper presented for review is very strongly focused on the Chinese economy, and more specifically on a particular group of enterprises operating in China. The article is written correctly both from the theoretical and methodological point of view. However, I would like to point out the following issues:

 1.     The Introduction and Abstract – The Authors did not clearly define the purpose(s) of the article. Thus, the element that combines the theoretical layer and the empirical layer is missing.

 2.     The greatest weakness of the paper, in my opinion, is the lack of a separate Discussion section. And I do not mean only distinguishing from the article a fragment of text that would be considered a discussion. I mean that the Authors of the article absolutely do not compare the results of their own research with findings previously presented in the literature. Discussion items appear in the Results and Discussion section and are valid. Nevertheless, I would expect a broader confrontation of research results with the findings of other authors dealing with the subject of Geographical Agglomeration, not only in relation to the Chinese economy. In its present form, it seems that the paper is not very attractive for foreign readers, who would particularly like to find content in the article that would indicate that the results of the study conducted by the Authors of the paper are to some extent similar or different from those presented by other authors in relation to other economies. Therefore, I would suggest making a separate section point where the suggested content is included.

 3.     Related to the argument described above is the issue of literature review. The literature review is extensive and current, but it is dominated by items relating to the Chinese economy. If the Authors thoroughly address the arguments described in point 2, they will naturally provide references to authors whose research also concerns other economies.

 In conclusion, I would like to say that there is absolutely nothing wrong with in-depth research on a specific area. However, it should be placed in the Discussion section in a broader context and confronted with research that covers other geographic areas. This greatly enhances the citation potential.  After making the suggested changes, the paper will be ready for publication, and I do not request a re-review.

Round 2

Reviewer 2 Report

The authors have incorporated the mentioned points in the revised manuscript. The paper can be published in its current form.